# Patient participant, healthcare professional, and stakeholder perspectives on the Pharmacy Homeless Outreach Engagement Non-medical Independent prescribing Rx (PHOENIx) community pharmacy pilot randomised controlled trial

Hannah Scobie[1], Shona MacKinnon[1], Karen Wood[1], Alessio Albanese[1], Yvonne Cunningham[1], Andrea Williamson[1], Jane Moir[2], Andrew McPherson[2], Cian Lombard[2], Steven Ross[3], Adnan Araf[4], Helena Heath[4], Richard Lowrie[5], Vibhu Paudyal[6]*, Frances Mair[1]

**1** General Practice & Primary Care, School of Health and Wellbeing, University of Glasgow, Glasgow, Scotland, **2** NHS Greater Glasgow & Clyde, Scotland, **3** Simon Community Scotland, Scotland, **4** Birmingham and Solihull Mental Health NHS Foundation Trust, Birmingham, England, **5** Centre for Homelessness and Inclusion Health, University of Edinburgh, Edinburgh, Scotland, **6** Florence Nightingale Faculty of Nursing, Midwifery and Palliative Care, King's College London, London, England

* vibhu.paudyal@kcl.ac.uk

## Abstract

### Background

People experiencing homelessness (PEH) face complex health and social care needs, contributing to poor health outcomes and premature mortality. The Pharmacy Homeless Outreach Engagement Non-medical Independent prescribing Rx (PHOENIx) intervention was developed to address these challenges through assertive outreach by NHS pharmacist independent prescribers working with third sector homelessness charity workers for PEH presenting to community pharmacy. This qualitative study aims to explore participant, healthcare professional, and stakeholder perceptions of the PHOENIx intervention and acceptability of trial procedures.

### Methods

Semi-structured interviews were conducted with trial participants, internal stakeholders (healthcare professionals/researchers), and external stakeholders (national and local) across two intervention sites (Glasgow and Birmingham) between March 2023 and February 2024. Data was analysed thematically using Normalisation Process Theory (NPT) as a conceptual framework.

**Data availability statement:** Data will be made available upon request. Data cannot be shared publicly because of conditions of the ethical approval received. By making the data publicly available we would be in breach of our ethical approval granted by NHS (REC reference 22/EM/0119). The ethics application, and therefore participant consent, state that anonymised data might be shared with qualified researchers but does not specify being made publicly available. As a result, the data cannot be made publicly available at this time. Ms. Fiona Hughes (Fiona.Hughes6@nhs.scot) can facilitate enquires and requests about data availability. Ms. Hughes is a research administrator working for NHS Scotland, and is not part of the research team.

**Funding:** This study was funded by the National Institute of Health and Care Research (NIHR) Health Services and Delivery Research scheme under commissioned call stream '20/56 Community Pharmacies'. [Grant award ID: NIHR133060]. Trial Sponsor: University of Birmingham, Birmingham, B15 2TT, United Kingdom. Views expressed are those of the author(s) and not necessarily those of the NIHR or the Department of Health and Social Care. Neither the study funder nor the sponsor had any role in the study design; collection, management, analysis, and interpretation of data; writing of the report; and the decision to submit the report for publication. Health Services and Delivery Research Programme NIHR133060, Vibhu Paudyal NIHR133060, Richard Lowrie.

**Competing interests:** The authors have declared that no competing interests exist.

## Results

Participants (n = 26; usual care n = 7; intervention n = 19) viewed PHOENIx as distinctively comprehensive, consistent, and caring, valuing its holistic approach and the trusting relationships developed with the team. Stakeholders (n = 16; internal n = 5; external n = 11) recognised PHOENIx's potential to fill gaps in current service provision for PEH, appreciating its flexible, outreach-based model. Challenges identified included resource constraints, integration with existing services, and concerns about long-term sustainability.

## Conclusions

The PHOENIx intervention shows promise in providing accessible, comprehensive healthcare which was acceptable to PEH. Its success in engaging this underserved population offers valuable lessons for service development. However, addressing challenges around resources, scalability, and sustainability will be important considerations for a future trial and wider implementation and expansion.

## 1. Introduction

Homelessness remains a significant and growing public health concern globally, with an estimated 895,000 people experiencing homelessness in Europe [1]. In Scotland, homeless applications increased from 35,759–39,006 in the year 2022–2023, representing a 9% increase [2]. Similarly, the rates of rough sleeping have also increased across other parts of the UK. In England there has been a 27% increase in rough sleeping in the year 2022–2023, as indicated by the most recent 'snapshot' [3].

People experiencing homelessness (PEH) face complex health and social care needs, contributing to premature mortality [4]. The average age of death for PEH is 45 years old, compared to 76 years in the general population [5]. Multiple factors contribute to this health inequality, including higher rates of physical and mental health conditions including substance use disorders, and barriers to accessing healthcare services [6,7].

PEH frequently experience multimorbidity, with a high prevalence of concurrent physical and mental health conditions [8]. Over 70% of PEH have underlying problem street-drug use [9]. Drug-related deaths among this population are particularly concerning, with a third of all deaths of PEH being street-drug-related [10]. Despite many receiving care from alcohol and drug recovery services, including daily prescribed opiate replacement therapy, there continues to be a high rate of overdoses experienced within the population [11].

Access to healthcare remains challenging for PEH due to stigma, discrimination, complex health systems that are difficult to navigate [12,13] and bureaucratic barriers [14,15]. As a result, unmet health needs are common [16,17]. Mental illness, pain, and substance dependency may be poorly managed [18], with evidence suggesting under-prescribing [11] and low rates of prescription adherence [19,20]. Patterns of

emergency healthcare utilisation differ between PEH and those who are housed, with PEH facing higher rates of emergency room visits, hospitalisations, and readmissions [21,22]. Reasons for these visits also differ, with PEH more likely to present for drug, alcohol, and mental health-related problems, often in need of acute and crisis care [21,22,23,24]. While the availability and uptake of preventative healthcare is lower among PEH [14,21,25], presentations to primary care tend to be reactive [26]. It is estimated that one-third of homeless deaths are preventable through timely and effective use of health services [27]. However, there is a lack of robustly evaluated person-centred, integrated, community-based effective and cost- effective complex interventions for PEH [28].

In response to these challenges, innovative approaches are needed to address the intersecting issues of homelessness and problem drug use [29]. The PHOENIx (Pharmacist led Homeless Outreach Engagement Non-medical Independent prescribing Rx) intervention was first developed as a novel, low-threshold service offered to PEH in Glasgow, Scotland [30–33], which was reported to be well received by PEH [34]. This complex health and social care intervention aimed to address the multiple unmet needs of PEH through weekly assertive outreach by National Health Service (NHS) pharmacist independent prescribers working in collaboration with third sector homelessness charity workers.

The PHOENIx intervention provided intensive and holistic health and social care support with persistent follow-up. Participants are visited via outreach and seen by PHOENIx workers (a pharmacist and third sector worker) at least once per week, with priorities set by the PEH and addressed gradually in succession. The intervention offered immediate health care and assessment, medication prescribing, and practical support such as advocacy, assistance with benefits applications, and provision of general wellbeing support [35].

Previous qualitative research based on the evaluation of the PHOENIx model of care in other settings has indicated that PHOENIx may improve health outcomes for PEH. Initial quantitative studies from previous evaluation of the PHOENIx model [11,36,37] have also highlighted the potential positive impact of the intervention on health-related quality of life [9], treatment burden, and delaying time to hospitalisation. However, there is a need for rigorous evaluation of whether the PHOENIx intervention can be delivered and how it is received, in other settings, e.g., community pharmacies and beyond Glasgow, in order to address the public health crisis of drug-related deaths and improve health outcomes for PEH.

This study aims to explore participant, healthcare professional, researcher and stakeholder perceptions of the PHOENIx intervention and acceptability of trial procedures related to the PHOENIx Community Pharmacy model in Glasgow (Scotland) and Birmingham (England), UK. By examining the experiences and perspectives of those involved in the intervention, we seek to inform future developments and potential scale-up of this complex intervention to address the major health inequalities faced by PEH in the community pharmacy setting.

## 2. Methods

### 2.1. Design

The study used a qualitative design, involving semi-structured interviews with PHOENIx trial patient participants, healthcare professionals (HCPs) and researchers involved in the intervention and key stakeholders, including HCPs and third sector workers. Interviews were conducted across the two trial sites (Glasgow and Birmingham) between March 2023 and February 2024.

### 2.2. Participants

Eligible participants were those who participated in the PHOENIx trial (both intervention and control arm participants) or played a role in the implementation of the intervention. Key stakeholders included those with an interest in the care/ service provision of PEH. Stakeholders were identified using a combination of professional networks, and snowball sampling. Ethical approval (REC reference 22/EM/0119; IRAS project ID: 309760) and Health Research Authority approval were obtained before recruitment and participants in the trial and qualitative process evaluation elements provided written informed consent before participation.

### 2.3. Procedures

**2.3.1. Trial participants.** All participants of the PHOENIx intervention, including those in the usual care group, were eligible to participate in interviews. Those eligible were approached by intervention workers to ascertain interest in participation. Those who expressed interest in taking part in interviews were supported by intervention staff and researchers to arrange a convenient time and date to conduct the interview. Interviews with intervention participants were conducted face-to-face, within community hubs that provide services to PEH at both trial sites. Those who expressed interest in participating were provided with an invitation pack containing a participant information sheet and consent form. At the time of interview, participants were given the opportunity to ask questions before providing written informed consent. Where needed, addition support was provided to potential participants to complete consent procedures, including the reading of the participant information sheet. Interviews with trial participants were around 60- minutes in duration (including consent procedures) and were audio-recorded. All interviews were transcribed verbatim, providing the data for qualitative analysis. Trial participants were given a £20 shopping voucher as a thank you for their time.

**2.3.2. Internal stakeholders (PHOENIx Intervention Staff and Researchers)/ External stakeholders (HCPs and third sector workers).** Those involved in the delivery of the PHOENIx intervention (intervention staff and researchers) and key external stakeholders were invited to participate in interviews on a voluntary basis. External stakeholders were identified using professional networks and 'snowball' sampling. Those who expressed an interest in taking part were contacted by email and provided with an invitation pack that included a participant information sheet and a copy of the consent form. Once agreement had been made, interviews were arranged at a convenient day and time for the interviewee. Interviews were conducted either face to face, or via online video platforms, whichever was most convenient for the participants. At the time of interview participants were given the opportunity to ask questions before providing written informed consent. Those interviewed remotely via a video platform were required to sign and email the consent form to the researcher. In addition, verbal consent was taken and recorded at the start of the interview. Interviews with staff delivering the PHOENIx intervention and researchers were between 30–60 minutes in duration and were audio recorded.

### 2.4. Development of interview schedule

A topic guide was used to structure the interviews. The topic guide, informed by Normalisation Process Theory (NPT), explored several aspects including overall perception of the PHOENIx intervention, engagement with the intervention team, perceived impacts of the intervention and the acceptability of the trial procedures. NPT is a sociological theory that describes four mechanisms that can be used to explain and evaluate the adoption and implementation of complex interventions or service innovations [36].

The topic guides for interviews with the PHOENIx team and other key stakeholders also included discussion topics such as the main challenges to achieving optimal health care and support for PEH and the barriers and facilitators of the intervention, which were not included in intervention participant interviews.

### 2.5. Data analysis

Data were analysed thematically with NPT used as an underpinning conceptual lens. The four constructs include: "coherence" (sense-making work); "cognitive participation" (relationship work); "collective action" (enacting work); and "reflexive monitoring" (appraisal work) [36]. These mechanisms were used to guide and structure conceptualisation of the analysis.

Transcripts were analysed thematically after initial familiarisation. Thematic analysis is a method for identifying, analysing, and reporting recurring patterns within data [38]. A two-step method of analyses was used, with the data initially analysed with the aim of identifying recurring themes, and then subsequently analysed using the NPT framework. This qualitative method of analysis allows for the use of both inductive (bottom-up) and deductive (top-down) approaches. This

'bottom up' approach allowed for greater initial exploration of the data, free from the constraints of a theoretical model. This methodology was considered to be appropriate in order to truly recognise themes that the theory might not capture [38].

An initial coding framework was developed after becoming familiar with the data. All transcripts were double-coded by HS, SM, KW and YC to ensure the robustness of the data interpretation and coding, with any queries or discrepancies discussed and resolved by the coding team. Following the double coding of all transcripts a summary report was produced, and themes generated mapped to the mechanisms described in the NPT framework.

## 3. Results

Our findings are presented under four key themes and components of the NPT framework:1) coherence; 2) cognitive participation; 3) collective action; and 4) reflexive monitoring. These themes are described for both intervention participants and stakeholders, and illustrative quotations are provided in the following section.

### 3.1. Intervention and Usual Care Participants

**3.1.1. Differentiating the intervention from usual care (Coherence).** Participants consistently distinguished the PHOENIx intervention from their usual care experiences. They highlighted the comprehensive and personalised nature of the service, which contrasted sharply with their previous encounters with health and social care services.

"Trying to find a way to move on from here, housing, medication, understanding your situation because everyone's unique. Most support workers, especially on this building, a support session to them is like:""Is your housing benefit still active? Great." You know what I mean? The Phoenix programme, they're actually trying to help find out a way to move forward: "Do you have bank accounts? Do you have ID? Do you have a housing option?" You speak to the support workers here and it's just like:"Have you paid your service charge? Okay, fuck off, you know." (IB05, Intervention participant)

"Yeah, kind of. Not lost in the system, just overlooked. Overlooked is… There's more important things to do. I think that's the way it feels like sometimes." (CB15, Usual care participant).

Many participants expressed frustration with their previous care, particularly in relation to addiction and mental health services. They felt that PHOENIx offered a more holistic and responsive approach.

"No because they're just, it's like they're just there to hand in my script. They [addictions team] do ask me: "How you feeling?" but most of the time I'm just in and back out." (IG011, Intervention participant)

"So I made an appointment for that, and I didn't go to that appointment, but they sent me another appointment out. […]And that's another reason why I have to see a doctor, because I've been discharged from the mental health team, because I haven't been good at keeping appointments with them either." (IB6, Intervention participant)

**3.1.2. Connection and consistency with PHOENIx team (Collective Action).** Participants valued the consistent and reliable nature of the PHOENIx intervention. The regular contact and follow-up were seen as crucial elements that distinguished this service from others they had experienced. This consistency helped to build trust and rapport between participants and the PHOENIx team, facilitating deeper engagement with the intervention.

"It's difficult to walk in and ask and saying what I need, because I'm used to doing everything myself but I wasn't doing it because I was so down. Once I knew how they were and that and they weren't judging, it was easy. They made it pleasurable. They made a good thing out of a bad situation." (IB1, Intervention participant)

"Well, building up a relationship, I'm getting to know them better and knowing that they are proper guys [PHOENIx team] that want to help you and not just, like, guys that are reading out of books and don't know nothing and they're trying but they've not got the heart in it, if you know what I mean?" (IG6, Intervention participant)

It was recognised by some intervention participants that the consistency of the support worker role was a challenge to the delivery of the intervention. This highlighted the need for consistency in this role.

"I want to continue with them, to be fair. I get a fair bit of help. It's obviously just unfortunate that my support worker has gone ill for the last two weeks, but I've been emailed about that. Obviously he's got some issues. I don't know if there's another support worker that steps in in the meanwhile." (IB5, Intervention participant)

**3.1.3. Holistic support provided by PHOENIx (Collective Action).** Participants recognised and appreciated both the practical and emotional support provided by the PHOENIx team. They noted how the intervention addressed not only their immediate health needs but also broader social and emotional concerns.

"Well, I spoke to the doctors I'm scared to go to the doctors, so whereas like, [support worker] absolutely amazing, comes and picks me up and took me to the doctors and stuff. I wouldn't have gone on my own so this project is absolutely amazing, it's getting me back on my feet and I wouldn't have even gone to the doctors. It's not because I can't go but I'm scared to go, you know when you're scared of the doctors you get a family member will drag…you know what it's like. I haven't got no one like that, so with this support project it gets me back on my feet, it just helps me…puts me in touch with my support services and just does what a family would do, because I haven't got a family like that. "(IB7, Intervention participant)

**3.1.4. Motivation to Engage with PHOENIx (Cognitive Participation).** The combination of practical support and emotional care provided by the PHOENIx team was crucial in sustaining participants' engagement with the intervention. Participants felt that the team's holistic approach addressed their complex needs in a way that encouraged ongoing participation. Trust and non-judgemental relationships were important to participants, with some participants feeling motivated by the trusting relationship with the PHOENIx team.

"It's been amazing, to be honest, it's been amazing. I've never had help like this and I've needed it for a long time and I've never had this help. Not only has he become my pharmacist, he's become a friend. We really have become friends and it's really good now. Someone to speak to you when I need to, you know what I mean? He's sorted out a lot of things for me as well, health-wise. I'm always in fights, and so if I get injured, I can go and see him and he looks after me. Do you know what I mean?" (IB3, Intervention participant).

**3.1.5. Work carried out by PHOENIx team (Collective Action).** Participants described a wide range of practical support provided by the PHOENIx team, including help with medication management, assistance with benefits applications, and support in accessing other services.

"Oh, no, it was easy with [pharmacist]. Just when I was explaining to him, and I think my doctor keeps putting me on the Salbutamol, the Symbicort and Becotide and [pharmacist] put me on, I'm only on one steroid, kind of thing.[…] And it's because they want me to sit and take them out of all the boxes and sit and take two puffs and then, shut one box and open another box. It's just a matter of taking it out the one box, taking two puffs and that, yes, a great difference." (IG3, Intervention participant)

"I'm getting somewhere now, I'm getting help with my benefits, I'm getting help with my housing, my prescriptions. And it's all thanks to going to the right sources, like the Phoenix team, for instance." (IG6, Intervention participant)

Alongside this practical support, participants emphasised the emotional work undertaken by the team. This included providing a listening ear, offering encouragement, and helping to build participants' confidence and self-esteem. Participants noted how this combination of practical and emotional support enabled them to engage more effectively with their care and make positive changes in their lives.

"I would say they've covered a lot with me, like my drink, my mental health, how I'm feeling this day, what am I eating, about my day. So they've covered a range of topics that no other people, even, like, my drugs worker has asked me, like what I'm eating and when I'm eating, am I eating regularly, like that." (IG11, Intervention participant)

### 3.1.6. Lack of power (Reflexive Monitoring).

Some participants expressed frustration that the PHOENIx team sometimes seemed limited in their ability to effect certain changes, particularly in areas like housing or specialist mental health support. Often, this lack of power is a result of the rules of other health and social care services, or long waiting times to access services. This perceived lack of power in certain areas led some participants to suggest that the team should be given more authority or resources to address complex needs.

"Aye. Well, I trusted their knowledge, but their knowledge didn't go any further than that. And I mean they've not got the powers to do anything. They can give me knowledge, but the knowledge that I then pass on to the ones that have got the power are just…" (IG2, Intervention participant)

"I'm still waiting on an appointment to meet an alcohol worker. See, they've actually been in contact with me before, but unbeknown to me, I've either been on the drink or, I'm very forgetful as well, like, you know? So I've either not remembered receiving the letter, and [support worker] has been in contact with them to say, look, they guy's still looking for a bit of support. But I'm still waiting to hear from them, like, you know, that's the [alcohol service]." (IG7, Intervention participant)

### 3.1.7. Commitment to long-term support (Reflexive Monitoring).

Participants consistently highlighted the importance of ongoing, long-term support. Many expressed concerns about what would happen when the intervention ended, indicating the value they placed on the continuous care provided by PHOENIx.

"Yeah, 100 per cent. I'll be sad if it stops, because who have I got to go to then? And I'll just go back to square one again." (IB3, Intervention participant)

This emphasis on long-term support reflects the complex and chronic nature of many participants' health and social issues, and their recognition of the need for sustained intervention. Usual care participants highlighted the reactive and inflexible nature of services which do not meet their needs.

"Because there isn't a lot of places, they always say:"Yes, have you been there?" but see when you go, they're either shut or there's nobody there or you need to come back another day. There's never somewhere where you can just do a pick up. I've only been told: "Right, yes, come up," or you can just walk in and it's there for you. It's always…because you can't have a breakdown at the correct time. Do you know what I'm talking about? If you need help, you can't make sure it's between nine and two or, do you know what I mean? It's just, when you need help, you need help." (CG13, Control participant)

**3.1.8. Acceptability of trial procedures (Reflexive Monitoring).** Intervention participants discussed the acceptability of the procedures of the intervention and trial. Overall, participants were supportive of the frequency and intensity of appointments with the PHOENIx team.

"I didn't think the Phoenix was going to help but I signed up to it, but I just thought it was just another one of the survey things. They're going to promise you the world and do nothing. They did as much as they have said." (IB5, Intervention participant)

However, there were challenges around the consistency of support from the support worker at one site as a result of recruitment and retention issues. For example, support workers moving on to other roles had an impact on the perceived consistency of support received.

Intervention participants indicated that the study questionnaire was too long in length and duration to complete, with some indicating that they did not believe the questions to be relevant and potentially stigmatising. A perceived lack of concentration and capability to answer some of the questions made the questionnaire challenging for some. Some indicated that the questionnaire was a perceived barrier to taking part in the study, while a significant number of participants did not remember having completed the study questionnaire.

"I can't really remember it to be honest but I ended up… It was a 20 minute thing and I ended up spending two hours." (IB2, Intervention participant)

"There were a few questions and I was like:"Forget that, let's go onto the…" yeah, you don't need to know that kind of thing, because why would you need to know certain things? It's got nothing to do with why I'm really here them questions I see as judgement. Someone else is going to look at this sheet of paper and see certain questions that are nothing to do with me but I'm going to be judged on these certain questions because of the answers." (IB3, Intervention participant)

## 3.2. Stakeholders

**3.2.1. Different roles within PHOENIx intervention (Coherence).** Stakeholders demonstrated a clear understanding of the various roles within the PHOENIx team and how these roles complemented each other to provide comprehensive care.

"I think that's good and I think it would get you so far. I mean, having a well-trained nurse practitioner, a pharmacist and a social worker, you would cover a lot of ground if those people were available to the person all the time. Obviously depending on how they all get on and everything. But those people working as like a mini- multidisciplinary team I think is a good idea." (BS11, External Stakeholder)

"Well, for me, it just seems like a really great way of doing things. Because from my perspective anyway, I [pharmacist] like to look at all the health stuff, and, yes, we…so, [support worker] is focused on the social things and the housing, but also, there's a bit of a grey area as well, this aspect of, social stuff can really solve mental health issues, or at least improve the whole kind of thing. So, we both work on both sides of it, if you see what I mean. And there's no real, I suppose…we just try and help people, whatever way is going to work." (GS21, Internal Stakeholder)

**3.2.2. Overall perception of PHOENIx intervention (Coherence).** Stakeholders generally held positive views of the intervention, recognising its potential to address gaps in current service provision for PEH. Providing anticipatory care was one such gap that was discussed.

"I think it's a really interesting project and it…from my experience with other homeless projects, it does bridge a gap that I could see and that I've seen acutely, where people, you cannot get them to the…you might be supporting them with their housing, but you cannot get them to the GP, or to A&E, or to any healthcare professional, for love nor money. And having someone that can…an actual floating prescriber is a game changer, really, for these people's health. And that…having that longer appointment time, and that sticky support, where they don't get cut off just because they haven't attended something, it's so vital, because we've got…the odds are stacked up against homeless people, so much that they need all the help that we can offer. And we've got these pharmacists all across the country that could do this. So, I'd like to see the project grow, definitely, and I'd like to be part of it." (BS9, Internal stakeholder)

"I think that we know there is a gap...well, sorry, I believe there is a gap around wound care for people in the city centre particularly, probably the whole of the city, that are not engaged in community services for whatever reason. That is potentially something that PHOENIx could support with. We know that there is ongoing, as there always has been, the issues into...pathways into mental health support are always more difficult. We have got decent processes on paper, but when you speak to service users then they don't reflect those processes. There is ongoing things like that that we are aware of and trying to do something about." (GS23, External stakeholder)

### 3.2.3. Challenges providing health care and support to PEH (Coherence).

Stakeholders acknowledged the complex challenges involved in providing care to PEH and saw PHOENIx as a potential solution to some of these issues. Some challenges highlighted by stakeholders was increased complexity and demand on services, lack of opportunity to engage with PEH, competing priorities of PEH and complex navigation of services.

"We have a really good system of care in the city. Alcohol and drug recovery services are really responsive. They hold a huge caseload, but the level of complexity and the level of demand is just always increasing. It is a really transient population in the city because it attracts people in from all over the country." (GS23, External stakeholder).

"So I think the main challenges are that people experiencing homelessness have lots of competing priorities, so they're not always able to prioritise their health. There's complications with their previous experience of using health services which are often very negative. They're used to being turned away from places, told that this isn't the right place for you, told you should be down the road, told you should have come before, you shouldn't come now, you should come later, all of those things. They're not really made to feel welcome in healthcare settings. I think that starts when people are young and they're not confident, they're often on their own, they often don't have support. That's the other thing, so they need support to navigate the health system. Usually we have that from friends, family, other people who will help us, but often people experiencing homelessness don't have that. Having support is very important and that can make a big difference." (BS11, External stakeholder)

### 3.2.4. External perception of PHOENIx team (Cognitive Participation).

Stakeholders outside the immediate PHOENIx team generally viewed the intervention positively, recognising the team's expertise and dedication. They appreciated the multidisciplinary nature of the team, the connection with the local community and viewed the team as highly skilled.

"My knowledge of it is that it was a really great team and delivered a really good service and was incredibly responsive. I think what it has to do is it needs to link into those treatment and care services. It can't just sit on its own, it can't just be its own thing. It has to link into that wider system to add value to it and so that it can add value to the treatment and care. I think if that was to happen it would be really, really great." (G23, External stakeholder)

"Again, it is probably like that where we are looking at the health needs through addictions. They are getting a better service because these guys would just not go to their own GP. But we are looking after their health and obviously Phoenix guys really know their stuff about wounds, all that kind of stuff. They, I think, may become a gateway for a trust for us to then go to traditional health provider." (GS19, External stakeholder)

**3.2.5. Perception of relationships between intervention team members (Cognitive Participation).** Internal stakeholders noted strong working relationships within the PHOENIx team, highlighting the importance of this collaboration for effective service delivery.

"Well, it's my job so aye, I love coming into my work, I love doing what I do and working with people who care, you know? So there's not a day…but I mean, we have some tough days in here, don't get me wrong, you know, we have some tough days. Aye. But there's good days as well, and no matter how tough a day, we can always make it better the next day. And working with people that are all aligned and on the same page, want to do the same thing, it's great. Aye." (GS22, Internal stakeholder)

**3.2.6. Perception of relationships between participants and PHOENIx team (Collective Action).** Stakeholders observed positive relationships developing between the PHOENIx team and participants, recognising this as a key factor in the intervention's success. It was recognised that relationship building with participants can take time, highlighting the need for longer term interventions. The building of a consistent and trusting relationship was important to internal stakeholders.

"I think that could do wonders really, because a lot of it is building the rapport and the relationship with somebody, getting that trust, you know, and if they know where to go they can support them. And you know, that trust is there, it's someone else that they could contact if they needed to." (BS14, External stakeholder)

"And that's the sort of service they needed. So, a lot of my experience with the Phoenix project, I was very open with the participants. I spoke to them very, like, was streetwise, sort of, speaking. I spoke to them like they were people. I wasn't, sort of, putting barriers up, I spoke to them like…you know, we had very open discussions. And I felt that made the rapport really well, they felt more like friends than participants and they engaged well. And that was my main aim." (BS11, Internal stakeholder)

**3.2.7. Challenges in provision of health care for homeless population or barriers to homeless care (Collective Action)**

Stakeholders identified numerous challenges in providing care to PEH, including data sharing between services, issues of engagement, continuity of care, and addressing complex, interconnected needs.

"Obviously we talk about things like data protection, those kinds of things all become a barrier because the sharing of information, because we struggle there all the time. As you can imagine, we've got relationships with the police, got relationships with enforcement teams in the city, and we've got relationships with the community mental health team, with the healthcare services and just the data sharing barrier is a constant thing that comes forward, so that's one barrier." (BS10, External stakeholder)

"I think obviously probably one of the biggest challenges is the fact that the access to care for people who are homeless is difficult. I think we've got to talk about it in the context of NHS care at the moment, access is difficult for

everybody. Just insofar as demands are very high, there isn't enough capacity for people. So if we're thinking about GP practices, at the moment to get an appointment there's a scramble at eight o'clock in the morning. You've got to be up, you've got to be on the phone, you've got to get through, there's a limited number of appointments to do it. So that very much relies on the fact that you've got a phone, it's working, you're awake, blah, blah, blah." (BS14, External stakeholder)

"I mean the first bit of what I say relates to what I've just said. I mean the health system's very rigid in lots of respects and it's not designed to deal with people who are facing homelessness or related issues. Most of the health service just isn't set up to cope with those people. As I said, how appointments are structured etcetera. So I think standard healthcare for most people who are homeless, isn't going to work for them the way it's set up at the moment. So I think actually we've got little opportunities." (BS14, External stakeholder)

### 3.9.8. Challenges of PHOENIx intervention identified during or after implementation or delivery (Collective Action).
While generally positive, stakeholders did identify some challenges in implementing PHOENIx, such as resource constraints and integration with existing services.

"Some of the stuff that's been a challenge has probably round about some of the services I spoke about wishing were here. Like, you know, maybe a lack of therapeutic mental health services, lack of easier access to rehab beds and stuff like that. So aye, that's probably it. Aye, probably, that's probably it." (GS22, Internal stakeholder)

Consistency of support provided at one intervention site was highlighted as a challenge. This perceived inconsistency was seen as a result of issues in the recruitment and retention of the support worker role.

"Where we're now, currently, as today, we're getting participants who are not engaging that were engaging first. And that's because there's been a break in the support side. But generally, obviously, there's only so much I can do. Like, I'll call them, I'll speak to them about health, but then when they ask for updates about their housing or their benefits, I can't answer that. That's because, you know, we've got that staggered, sort of, break down. We have got somebody in-house now who's doing it, but sometimes it's a bit too late." (BS11, Internal stakeholder)

The workload of the intervention team was also identified as a challenge, with high caseloads to manage, alongside administration for both intervention participants and the intervention itself.

"Yes, workload is heavy, aye, workload is heavy. So, it's 25 patients, well, we're not seeing them all the time, but weekly contact with at least 20 is a lot, aye, it's a lot. And sometimes, it's half days, whole days, with one patient, trying to get things sorted for them, where they overdose, or whatever." (GS21, Internal stakeholder)

"No. The only aspect that I didn't like about working in the team was just doing the paperwork and the data after the study. Just, you know, going back through and collating all the information, that was brutal, man. Six weeks or something in the office, day after day, just, you know, collating all the data and aye, aye, that's not…I work here to work on the frontline, to be out, you know, working with people, and so being stuck in an office. But that's the only thing I've not liked about working with the team." (GS22, Internal stakeholder)

### 3.2.9. Optimal care for homeless population (Coherence).
Stakeholders viewed PHOENIx as aligning closely with what they considered optimal care for PEH, particularly in its holistic, person-centred approach. Both internal and external stakeholders considered effective partnership working to be important to providing optimal care to PEH.

"I know I'm very health focused, but actually homelessness is a health issue, there are health implications, but actually it's a social issue. Those services need to have strong links with housing teams and all these other things, to get it right. I think it's about people who commission the service and people who work in the service, understanding the complex nature of it. Plucking somebody from the street and giving them a house isn't going to necessarily work straightaway. Some people haven't lived in a house for a long time. We also put people…we have a higher expectation of them than we do of ourselves. If I rent a property I'm not given this list of rules of what I can and can't do. I'm not scrutinised. Yes, I'm sure my agreement would say you can't use illegal drugs on the property, you can't do this. But no-one's going to come and check." (BS14, External stakeholder)

"For optimal care. They need a holistic approach. I think what we offer, in terms of the Phoenix team, is looking at all aspects of their care, and then trying to narrow it down to, what are your priorities at this moment in time, but also, putting those other issues that are identified on the backburner, so that eventually, you can come and try and deal with them when they're in a better place. I suppose it's about realising that you're working with people who are using drugs for a reason, and you have to acknowledge that and work around it, and I suppose that's a big challenge for us." (GS21, Internal Stakeholder)

### 3.2.10. Role of PHOENIx (Collective Action).
Stakeholders saw PHOENIx as playing a unique and valuable role in the care landscape for PEH, bridging gaps between different services and providing continuity of care. Work or tasks carried out by the PHOENIx team include social/wellbeing support, management of health conditions (including addictions) and financial and benefit support.

"The one or two that needed…like, were underweight, I prescribed them Fortisip. But other than that, we've not really had to do anything. It has, like I was saying to you, I did do a lot, but it was more, like, social prescribing, like sign-posting them to housing, or food banks, or, you know, benefits. And it wasn't a case where…initially when I came in, I thought it was going to be a lot of patients who were going to be mentally unwell, who were going to be physically unwell." (BS11, Internal Stakeholder)

The PHOENIx team also provided support with navigating and referring to other services that might be perceived to be previously too complex for those with significant and complex needs.

"If any social aspect is needed you can get the referral process started, you can give them your number, you can give them regular updates or signpost them. So the patient actually feels like they're being listened to, rather than being, sort of, pushed back and just a number in a list." (BS11, Internal stakeholder)

### 3.9.11. Positive features of PHOENIx intervention identified during and after implementation (Reflexive Monitoring).
Stakeholders highlighted several positive aspects of PHOENIx, including its flexibility, holistic approach, and association with well-known or trusted organisations.

"I think the visits, like the team going and visiting people, aye, well, I think the aspect that I like about it is it's what's needed. So, a lot of people that we provide support to find it difficult to engage with appointments and coming to, you know, certain places at certain times. So the team goes there. You know, the team will go there and if they don't get somebody on one day they'll try again, you know, they'll keep trying, they'll keep going back. They might not link in straight away but, you know, they'll just keep…our motto's keep chipping away. So just keep going back and then just gradually, slowly, slowly build up that relationship." (GS22, Internal stakeholder)

**3.2.12.  Suggested changes for future trial of PHOENIx intervention (Reflexive Monitoring).**  While generally positive, stakeholders did suggest some potential improvements for future iterations of PHOENIx. Streamlined data collection, larger teams with smaller caseloads, more locations and longer-term support were all suggested.
Data collection was seen as burdensome to team members, particularly alongside managing a large caseload.

"I also think that we really need to look at what data we're collecting. Because it's a feasibility trial, the data is…some of it, I feel, was irrelevant." (BS9, Internal stakeholder)

"So, I don't really know what the optimal caseload…and it would probably vary depending on what was going on for, you know, different people within that caseload. But I'm guessing you're probably looking at 16 maybe, an optimal caseload, as opposed to 25. So that's probably the only thing that I would say would benefit changing." (GS21, Internal stakeholder)

Stakeholders agreed that community pharmacy was an appropriate setting for the intervention, however, some stakeholders suggested different geographical locations, including those outside of city centres to reduce geographical barriers.

"I think that Phoenix research needs to be done in a few locations. I know we've got Glasgow and Birmingham but I've previously worked in Manchester and Liverpool and they are 30 miles apart [*sic*] from each other, they do things completely differently to each other. I think it might have changed slightly now. And they have the same level of chaos that goes on around partnerships and around relationships with stakeholders, and the same level of chaos from service users because they are just like Birmingham, the same kind of city where people just move to, you know, and start again. Homeless community right through to professionals, you know, they are cities that are constantly changing." (BS10, External stakeholder)

"So I think just different pockets of the country where you've got those big urban cities like Manchester, Liverpool, Glasgow and Birmingham, versus some of those more rural Devon, Gloucestershire kind of scenarios, and Worcestershire as well, there's a massive homeless problem in Worcestershire but it's all quite hidden, and I think from the perspective of research, it might be beneficial to tap into some of those areas and just see the disparity and the differences between town to town, city to city." (BS10, External stakeholder)

Longer-term support was also suggested for a future trial of PHOENIx.

"Maybe we just need longer, we need longer time. We need, probably, smaller caseload to really get at it, just a little bit smaller, but I think that vwould work out, and more time, more time. If we had longer time for the thing, I'm sure we could see different…maybe slightly different results […]" (GS21, Internal stakeholder)

**3.9.13.  Barriers to future trial or implementation of PHOENIx intervention (Reflexive Monitoring).**  Stakeholders identified potential barriers to scaling up PHOENIx, including resource constraints, support and buy-in from policy makers, and intervention staffing.
Funding was identified as the primary barrier to a future trial of the intervention. Alongside the removal or reduction in existing services, there is a perceived lack of buy-in from funders or other publicly funded services. Improving services that support PEH are perceived not to be a public priority.

"I think it's going to be funding. Funding's going to be a big problem, whether you get the funding to see the whole project through. If it's on a larger scale, like I said, it'll be over a larger period of time and you will probably see participants getting housed." (BS11, Internal stakeholder)

"The Government aren't really invested in tackling the issue, because at the heart of all, or nearly all health inequalities, is poverty. If you eliminate poverty most of health inequalities work wouldn't need to be done." (BS14, External stakeholder)

The staffing of the intervention roles is also seen as a barrier to future implementation. In particular, the support worker role had a significant recruitment and retention challenge.

"It's money, it's always money. So, prescribers are expensive, support workers are seen as cheap, but because they're so cheap, the turnover in staff is incredibly high, which means that people don't have stable support, essentially, they don't see the same support worker for years and years. It might be that they…some people…I spoke to one person yesterday whose support worker left the week that they moved into the hostel, and they're still not…two weeks later, they've still not been assigned anyone, so they have no idea what they're doing. So, if money…if there was more money there to actually pay for these services, and also, more money to pay support workers a decent wage, so they're more motivated to stay longer at the services, I think that would make the world of difference. The cost of having someone who can prescribe is…I mean, I….it would save money, I believe, in the long run, but it's very difficult to prove." (BS9, Internal stakeholder).

## 4. Discussion

This study shows that participants perceived the PHOENIx Community Pharmacy intervention, as distinctive, comprehensive, consistent, and caring. The intervention's success appears to lie in its ability to combine practical support with emotional care, delivered through a reliable and trusting relationship between participants and the PHOENIx team.

Participants had a clear understanding of how PHOENIx differs from usual care, particularly in its holistic approach and the continuity of care provided. Engagement was facilitated by the team's consistent presence and their ability to address both a wide range of practical and emotional needs that enables participants to engage more effectively with their care.

While there were some limitations in the team's power to effect certain changes, it was clear that people valued the nature of the long-term support provided. Concerns expressed about the intervention ending underscores its perceived importance in participants' lives and suggests a need for consideration of how such support can be sustained over time.

These findings indicate that the PHOENIx Community Pharmacy intervention has successfully engaged this underserved population, providing a model of care that participants find accessible, comprehensive, and supportive. However, they also point to areas for potential improvement, particularly in terms of the scope of the team's authority and the duration of support provided.

Stakeholders' perspectives on the PHOENIx Community Pharmacy intervention reveal a generally positive view of the intervention's potential to improve care for PEH. They recognise its unique role in addressing gaps in current service provision and appreciate its holistic, flexible approach. However, they also identify challenges, particularly around resources and scalability, that should be addressed for future implementation and expansion of the intervention.

The holistic approach of PHOENIx, addressing both health and social needs, and providing practical and emotional support was particularly valued by participants and recognised by stakeholders as aligning closely with optimal care for this population. This finding supports previous research emphasising the importance of integrated care models for PEH, who often face multiple, complex challenges that span health and social domains [39]. Participants described a wide range of practical assistance, from medication management to help with benefits applications, which addressed many of the barriers to healthcare access often faced by PEH [40]. Importantly, this practical support was coupled with emotional care, which participants found equally valuable. This holistic approach aligns with calls for trauma-informed practice for PEH, recognising the complex interplay between physical health, mental health, and social circumstances in this

population [40,41]. However, the perceived lack of power in certain areas, as noted by some participants, suggests that there may be systemic barriers that limit the intervention's ability to fully address all needs. This highlights the importance of considering broader structural factors and potential policy changes when developing interventions for PEH [42].

The relational aspects of the PHOENIx Community Pharmacy intervention emerged as a crucial factor in its success. Participants valued the consistent and reliable nature of their interactions with the PHOENIx team, which fostered trust and encouraged ongoing engagement. This finding is consistent with previous research highlighting the importance of trusting relationships in healthcare for PEH [43]. Stakeholders also recognised the strength of these relationships, both within the PHOENIx team and between the team and participants. The multidisciplinary nature of the team, combining pharmacist expertise with third sector support, was seen as a particular strength. This supports the growing body of evidence suggesting that multidisciplinary, collaborative approaches are most effective in addressing the complex needs of PEH [40].

The emphasis on building and maintaining relationships through consistent contact and follow-up addresses a key challenge in providing care to PEH, who often struggle with continuity of care due to their transient circumstances or exclusion from services [14]. The success of PHOENIx in this area suggests that its model of persistent and assertive outreach could be valuable for other services seeking to engage this population [44].

Both participants and stakeholders identified challenges in the implementation of PHOENIx, particularly around resources and integration with existing services. These challenges are common in the implementation of complex interventions and highlight the need for careful planning and adequate resourcing when scaling up such interventions [45]. Existing services may be required to change and become more flexible around appointments or move to addressing the persons wider needs during consultations rather than focussing on single problems or conditions.

Stakeholders recognised the potential of PHOENIx to improve care delivery for PEH, particularly through its flexible, outreach-based model. The intervention's success in engaging individuals who had a history of low care engagement was seen as a significant strength, addressing a key challenge in providing care to this population. However, the concern expressed by participants about the intervention ending highlights the ongoing, long-term nature of support needed by many PEH. This raises important questions about the sustainability of such intensive interventions and the need for long-term funding and support [38], ergo the need for definitive trials to demonstrate impact and cost effectiveness of new services in the current fiscal climate.

### 4.1. Considerations for future RCTs

Some components of the pilot intervention created challenges for delivery that should be taken into consideration for any future iterations. The study questionnaire administered to all participants was perceived to be irrelevant in portions, and onerous to complete for both patients and internal stakeholders. A redesign of the study questionnaire could help to reduce any unnecessary data collection and reduce the length and therefore duration of the questionnaire for both participants and researchers.

Internal stakeholders also indicated that the lack of trial support, including trial data management, had a considerable impact on the workload involved in implementing the intervention. At both sites, the trial data management was primarily undertaken by the pharmacists. Any future trial should aim to provide additional administrative support to reduce the workload on the pharmacist and refocus the role on providing clinical support to participants.

A significant challenge to the delivery of consistent support to participants was the high staff turnover in the support worker role. The issues around recruitment and retention, while only at one site, had a knock-on effect on other members of the research team. Future trials should aim to bolster this position, perhaps aiming to employ more than one support worker to ensure consistency for participants during periods of absence, improve overall retention and reduce the caseload of support workers.

Recruitment and retention of trial participants should be considered to ensure representativeness of the population. PHOENIx participants represent 0.6% of Glasgow's homeless population [46], and 0.3% of PEH in Birmingham [47]. A full RCT should aim to improve recruitment of study participants. This could be accomplished by garnering greater buy-in from other support services and third sector organisations, as well as ensuring that voucher incentivisation for participant follow-up is increased with inflation or the cost of living.

### 4.2. Implications for practice and policy

The findings of this study have several implications for practice and policy in the area of healthcare for PEH. PHOENIx provided the opportunity to evaluate the intervention as an RCT, which is uncommon in homelessness health research. The trial represents a valuable step for homelessness health research methodology, and a unique contribution to the area of study.

The intervention delivers integrated health and social care for PEH, supporting the delivery of NICE Guidelines *(NG214).* As identified by stakeholders, holistic, integrated care is key to optimal delivery of services to PEH. The success of PHOENIx's comprehensive approach supports the need for services that address both generalist health and social needs in an integrated manner. Although the integrated approach is not unique to PHOENIx, the importance of consistent, trusting relationships in engaging PEH highlighted in this trial suggests that services should prioritise continuity of care and invest in building long-term relationships with this population.

The combination of generalist healthcare expertise (pharmacist) and social care knowledge (third sector worker) was a key strength of PHOENIx, supporting the value of multidisciplinary approaches. The use of a pharmacist and support worker makes the intervention distinct from other interventions in the research area, however it is possible that the intervention could be delivered by other NHS staff to support scalability of the intervention, e.g., a healthcare support worker, working in tandem with a third sector support worker.

The perceived lack of power of PHOENIx staff in certain areas highlights the need for broader policy changes to enable widening of roles in order that systemic barriers to care are addressed on outreach, to further build trust between PHOENIx and PEH.

### 4.3. Limitations and future research

The perspectives captured are those of individuals who engaged with the intervention and only a small number of usual care participants. Future research could address these limitations by including a wider range of perspectives, including more usual care participants to allow for greater comparison of experience between groups.

Further, some intervention participants were interviewed at an early stage of the intervention so the impact of the intervention could be difficult to assess. Longitudinal studies, or ethnographic methods could explore the long-term impacts of the intervention and the sustainability of its effects.

## 5. Conclusion

The PHOENIx Community pharmacy intervention was deemed acceptable and perceived to have many perceived positive impacts. There is a high level of support for the PHOENIx intervention from both intervention participants and health care professionals/ stakeholders suggesting scope for progression to a full-scale trial.

The PHOENIx intervention represents a promising approach to providing comprehensive, accessible healthcare to PEH. Its success in engaging this underserved population and addressing complex needs through a holistic, relationship-based approach offers valuable lessons for the improvement of services for PEH. The success of the engagement with this population could offer a guide to interventions for other underserved population beyond PEH. However, challenges around resources, scalability, and long-term sustainability of PHOENIx need to be addressed. As healthcare

systems struggle with the challenge of complex needs of PEH, and recruitment crises among clinical professionals, models such as PHOENIx offer a template for more effective, person-centred care that holds promise for improving health outcomes for this vulnerable population.

## Author contributions

**Conceptualization:** Andrea Williamson, Richard Lowrie, Vibhu Paudyal, Frances Mair.

**Data curation:** Jane Moir, Andrew McPherson, Adnan Araf, Helena Heath.

**Formal analysis:** Hannah Scobie, Shona MacKinnon, Karen Wood, Yvonne Cunningham.

**Funding acquisition:** Andrea Williamson, Richard Lowrie, Vibhu Paudyal, Frances Mair.

**Investigation:** Hannah Scobie, Shona MacKinnon, Alessio Albanese, Helena Heath.

**Methodology:** Richard Lowrie, Vibhu Paudyal, Frances Mair.

**Project administration:** Alessio Albanese, Jane Moir, Andrew McPherson, Adnan Araf, Helena Heath.

**Resources:** Jane Moir, Andrew McPherson, Cian Lombard, Steven Ross, Adnan Araf, Helena Heath.

**Supervision:** Andrea Williamson, Richard Lowrie, Vibhu Paudyal, Frances Mair.

**Writing – original draft:** Hannah Scobie, Frances Mair.

**Writing – review & editing:** Hannah Scobie, Shona MacKinnon, Karen Wood, Alessio Albanese, Yvonne Cunningham, Andrea Williamson, Andrew McPherson, Richard Lowrie, Vibhu Paudyal, Frances Mair.

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
