## [Decision Letter · Decision Letter 0]

29 Jan 2025

PONE-D-24-52416Patient participant, healthcare professional, and stakeholder perspectives on the Pharmacy Homeless Outreach Engagement Non-medical Independent prescribing Rx (PHOENIx) community pharmacy pilot randomised controlled trialPLOS ONE

Dear Dr. Scobie,

Thank you for submitting your manuscript to PLOS ONE. After careful consideration, we feel that it has merit but does not fully meet PLOS ONE’s publication criteria as it currently stands. Therefore, we invite you to submit a revised version of the manuscript that addresses the points raised during the review process.

We look forward to receiving your revised manuscript.

Kind regards,

Ilhem Berrou, PhD

Academic Editor

PLOS ONE

Journal Requirements:

2. In the online submission form, you indicated that data will be made available upon request. Data cannot be shared publicly because of conditions of the ethical approval received.

Reviewers' comments:

Reviewer's Responses to Questions

**Comments to the Author**

1. Is the manuscript technically sound, and do the data support the conclusions?

Reviewer #1: Yes

Reviewer #2: Yes

2. Has the statistical analysis been performed appropriately and rigorously? 

Reviewer #1: Yes

Reviewer #2: Yes

3. Have the authors made all data underlying the findings in their manuscript fully available?

Reviewer #1: Yes

Reviewer #2: Yes

4. Is the manuscript presented in an intelligible fashion and written in standard English?

Reviewer #1: Yes

Reviewer #2: Yes

5. Review Comments to the Author

Reviewer #1: Few points to consider her:

First: There are few repetitive content in the introduction and results sections. You may try to avoid that to avoid distracting the reader of your manuscript.

Also, please emphasize the unique contributions of your study, particularly its implications for scaling the PHOENIx model.

Lastly, in your conclusion, you may consider discussing how findings can guide interventions for other underserved populations beyond PEH.

Reviewer #2: This is a well written manuscript about people experiencing homelessness (PEH) and using normalization Process Theory (NPT) for thematic analysis. The NPT was based on four constructs including coherence, cognitive participation, collective participation and reflexive. Participants signed a consent form before participating to the study. The conclusion supports the findings of this study.

The weakness of the study was a very small sample size. I have the following questions and comments.

(1) What percentage of PEH represent the sample size for the cities of Glasgow and Birmingham?

(2) The authors need to address in the discussion section what strategies they will use to increase the number of PEH participants for future studies?

6. PLOS authors have the option to publish the peer review history of their article (what does this mean? ). If published, this will include your full peer review and any attached files.

**Do you want your identity to be public for this peer review?** For information about this choice, including consent withdrawal, please see our Privacy Policy .

Reviewer #1: No

Reviewer #2: **Yes: ** Albert Nguessan Ngo

---

## [Author Response · Author response to Decision Letter 1]

3 Apr 2025

Dear Editor/ Reviewer,

Thank you for your considered and helpful comments on the manuscript entitled: ‘Patient participant, healthcare professional, and stakeholder perspectives on the Pharmacy Homeless Outreach Engagement Non-medical Independent prescribing Rx (PHOENIx) community pharmacy pilot randomised controlled trial’.

The authors have reviewed the comments provided and have made amendments where appropriate. A table of responses to comments can be found below:

Table of reviewer comments and response from authors

Reviewer Comment Response

1 There are few repetitive content in the introduction and results sections. You may try to avoid that to avoid distracting the reader of your manuscript. The text was checked for clarity and consistency.

1 Please emphasize the unique contributions of your study, particularly its implications for scaling the PHOENIx model.

Text added to the discussion that aims to highlight the unique contribution to the field of study, as well as implications/ suggestions for scalability.

1 In your conclusion, you may consider discussing how findings can guide interventions for other underserved populations beyond PEH.

Text has been added to the conclusion to highlight how successful engagement of an underserved population (PEH) can be used as a guide for future interventions looking to take a person-centred approach with other marginalised groups e.g. migrant population.

2 What percentage of PEH represent the sample size for the cities of Glasgow and Birmingham?

This has been noted, with appropriate text added to ‘Considerations for Future RCT’ section within the discussion.

2 The authors need to address in the discussion section what strategies they will use to increase the number of PEH participants for future studies?

This has been noted, with appropriate text added to ‘Considerations for Future RCT’ section within the discussion. This includes suggestions for improving recruitment and retention of study participants.

Other changes that have been made to the manuscript include:

- Amended formatting to meet PLOS One submission guidelines, including correct heading formatting.

- An addition of two further references, added in response to reviewer twos request for figures relating to percentage of number of PEH in Glasgow and Birmingham.

- All references have been checked, and there are no articles that have been retracted at this time.

In relation to the data availability statement, as highlighted by the editor, we would ask to request an exemption on the grounds that by making the data available we would be in breach of our ethical approval granted by NHS (REC reference 22/EM/0119). The ethics application, and therefore participant consent, state that anonymised data might be shared with qualified researchers but does not specify being made publicly available. As a result, the data cannot be made publicly available at this time. If required, the approved ethics application can be provided.

Best wishes,

Hannah Scobie

---

## [Editor Report · Decision Letter 1]

29 Apr 2025

Patient participant, healthcare professional, and stakeholder perspectives on the Pharmacy Homeless Outreach Engagement Non-medical Independent prescribing Rx (PHOENIx) community pharmacy pilot randomised controlled trial

PONE-D-24-52416R1

Dear Dr. Scobie,

We’re pleased to inform you that your manuscript has been judged scientifically suitable for publication and will be formally accepted for publication once it meets all outstanding technical requirements.

Kind regards,

Ilhem Berrou, PhD

Academic Editor

PLOS ONE
---

## [Editor Report · Acceptance letter]

PONE-D-24-52416R1

PLOS ONE

Dear Dr. Scobie,

I'm pleased to inform you that your manuscript has been deemed suitable for publication in PLOS ONE. Congratulations! Your manuscript is now being handed over to our production team.

Kind regards,

on behalf of

Dr. Ilhem Berrou

Academic Editor

PLOS ONE